# SARS-CoV-2 M^pro^: A Potential Target for Peptidomimetics and Small-Molecule Inhibitors

**DOI:** 10.3390/biom11040607

**Published:** 2021-04-19

**Authors:** Andrea Citarella, Angela Scala, Anna Piperno, Nicola Micale

**Affiliations:** Department of Chemical, Biological, Pharmaceutical, and Environmental Sciences, University of Messina, 98166 Messina, Italy; acitarella@unimore.it (A.C.); ascala@unime.it (A.S.); apiperno@unime.it (A.P.)

**Keywords:** COVID-19, SARS-CoV-2 M^pro^, protease inhibitors, coronavirus, peptidomimetics, remdesivir

## Abstract

The uncontrolled spread of the COVID-19 pandemic caused by the new coronavirus SARS-CoV-2 during 2020–2021 is one of the most devastating events in the history, with remarkable impacts on the health, economic systems, and habits of the entire world population. While some effective vaccines are nowadays approved and extensively administered, the long-term efficacy and safety of this line of intervention is constantly under debate as coronaviruses rapidly mutate and several SARS-CoV-2 variants have been already identified worldwide. Then, the WHO’s main recommendations to prevent severe clinical complications by COVID-19 are still essentially based on social distancing and limitation of human interactions, therefore the identification of new target-based drugs became a priority. Several strategies have been proposed to counteract such viral infection, including the repurposing of FDA already approved for the treatment of HIV, HCV, and EBOLA, inter alia. Among the evaluated compounds, inhibitors of the main protease of the coronavirus (M^pro^) are becoming more and more promising candidates. M^pro^ holds a pivotal role during the onset of the infection and its function is intimately related with the beginning of viral replication. The interruption of its catalytic activity could represent a relevant strategy for the development of anti-coronavirus drugs. SARS-CoV-2 M^pro^ is a peculiar cysteine protease of the coronavirus family, responsible for the replication and infectivity of the parasite. This review offers a detailed analysis of the repurposed drugs and the newly synthesized molecules developed to date for the treatment of COVID-19 which share the common feature of targeting SARS-CoV-2 M^pro^, as well as a brief overview of the main enzymatic and cell-based assays to efficaciously screen such compounds.

## 1. Introduction

Prior to 2003, the only known pathogenic coronaviruses were hCoV-229E and hCoV-OC43, both responsible for trivial respiratory diseases with symptoms similar to the common cold [1]. The discovery of Severe Acute Respiratory Syndrome coronavirus (SARS-CoV) and Middle East Respiratory Syndrome coronavirus which caused the SARS (2003) and MERS (2012) outbreak, respectively, have raised enormous concerns about the dangerousness and high contagiousness of the new coronaviruses [2,3]. The rapid worldwide spread of the new coronavirus disease 2019 (COVID-19) has put global public health in serious difficulty since the beginning of 2020. COVID-19 is a respiratory disease, affecting mostly the lungs and causing symptoms of varying degrees of morbidity [4]. To date, this infection caused by new coronavirus SARS-CoV-2 affected more than 120 million people all over the world—~30 million in the US and over 3 million in Italy—causing a high number of deaths [5]. Despite a global ongoing large-scale vaccination, the problems associated with this viral disease are far from being solved, in particular due to insurgence and diffusion of new strains and, in some cases, inadequacy of health services. Therefore, the need to find new therapeutic strategies for the prompt, effective, and affordable treatment of this dangerous infection is becoming crucial.

The causative agent of COVID-19 is SARS-CoV-2, a single stranded RNA virus (+ ssRNA) belonging to the *Coronaviridae* family capable of spreading among humans and animals [6]. The genome of SARS-CoV-2 is represented by two open reading frames which encodes two large polyproteins, pp1a and pp1ab. Following proteolytic cleavage, these are translated into mature non-structural proteins (NSPs). The enzymes involved in this conversion are two cysteine proteases, namely the papain-like protease PL^pro^ and the chymotrypsin-like protease 3CL^pro^, the latter more commonly known as M^pro^. They are indispensable for virus proliferation and infectivity, as they allow the maturation of viral polyproteins and, consequently, their assembly into new virions. It is commonly believed that finding specific new therapies aimed at inhibiting these proteolytic processes can be of great use in the fight against COVID-19, as well as in the prevention of subsequent viral epidemics.

The replication cycle of SARS-CoV-2 is depicted in Figure 1. The entry of the virus into the lung cell is regulated by the interaction of the spike protein (S protein) with the angiotensin-converting enzyme 2 (ACE-2) receptor, located on the surface of the pneumocyte [7]. The S protein is first processed by a proteolytic cleavage involving the S1/S2 (or S2 ‘) sites from a serine protease located on the surface of the human cell, called TMPRSS-2 [8]. TMPRSS-2 belongs to a family of transmembrane serine proteases and it is essentially involved in the entrance of the virus into the host cell. The proteolytic cleavage carried out by TMPRSS-2 on the viral S protein allows the exposure of the binding site of the external surface of the virus to the ACE-2 receptor, enabling the entry. It is noteworthy that the affinity of the S protein of SARS-CoV-2 for the ACE-2 receptor is about 10 times greater than that of SARS-CoV [9]. Thus, the virus penetrates the host cell by receptor-mediated endocytosis or fusion with the plasma membrane [10]. Hoffmann M. et al. demonstrated how camostat, a TMPRSS2 inhibitor containing a terminal guanidine endowed with anti-inflammatory and antifibrotic properties, can block SARS-CoV-2 infection in lung cells [8], suggesting that targeting this protease may be one of the potential strategies to discover and develop novel anti-COVID-19 fusion inhibitors [11].

Various important proteolytic enzymes are involved in the replication and infectious capacity of SARS-CoV-2, including SARS-CoV-2 M^pro^, SARS-CoV-2 PL^pro^, and TMPRSS-2. However, only molecules capable of selectively inhibiting the SARS-CoV-2 M^pro^ activity have been effectively developed. Despite their important role, the other two viral proteolytic enzymes are difficult to target due to their high degree of similarity with human proteases and the consequent drawbacks related to phenomena of non-specific inhibition. Therefore, this review is focused on the structure of the SARS-CoV-2 M^pro^ and the discovery of its inhibitors (repurposed drugs or newly synthesized compounds) for the treatment of COVID-19. Furthermore, the main enzymatic assays and cell-based methods for their biological assessments will be briefly discussed.

## 2. Structure of SARS-CoV-2 Main Protease (M^pro^)

As above-stated, SARS-CoV-2 M^pro^ is certainly the most interesting molecular target for a pharmacological treatment of COVID-19 in view of its highly specific structure [12,13,14]. SARS-CoV-2 M^pro^ (Figure 2) plays a leading role in the replication cycle of all coronaviruses [15]. In association with the second fundamental viral protease SARS-CoV-2 PL^pro^, it performs more than 11 cuts at the 1a/1ab polyprotein (pp) and hydrolyzes it in 16 mature non-structural proteins (NSPS) [16,17,18,19]. The latter are involved in the synthesis of subgenomic RNA necessary for the production, by the host cell, of viral structural proteins (i.e., envelope, membrane, spike, and nucleocapsid). M^pro^ is widely conserved among coronaviruses and it possesses a high degree of homology with that of other members of the same family such as MERS-CoV and SARS-CoV M^pro^. On March 2020, Zhang L. et al. published the X-ray crystallographic structure (PDB 6lu7) of SARS-CoV-2 M^pro^ bound to a peptide-based Michael acceptor inhibitor known as N3 (see herein after) [12], previously designed as a wide-spectrum inhibitor of coronavirus M^pro^. Thank to this early outcome it was possible to put in evidence and compare the structural analogy between SARS-CoV M^pro^ and SARS-CoV-2 M^pro^, creating a library of compounds as potential lead structures for these targets [20]. Since affinity model studies have revealed a homology of 96% between these two M^pro^, it is reasonable to hypothesize that inhibitors designed for SARS-CoV M^pro^ may be suitable inhibitors also for the type 2 M^pro^ [21]. Although only 12 amino acid residues differ between the first and the second type of M^pro^, this causes significant differences in the catalytic performance of the protease and the inhibitory activity of molecules [22]. In particular, only one of the different residues (Ser46) is located in proximity of the catalytic site and it is probably the one influencing most of the proteolytic activity. This residue is placed in a Cys44-Pro52 loop, which regulates the entry of solvation molecules and inhibitors into the enzymatic cavity. The rigidity of this loop is greater in SARS-CoV-2 M^pro^, and this significantly reduces the capability of the inhibitor to reach the binding site [22]. Indeed, it has also been observed that the width of the external pockets changes between the two types of virus—SARS-CoV M^pro^ has a larger external pocket—and probably this could also contribute to reduce the rate of the drug entry into the enzyme cavity [23]. Despite the great similarity found between SARS-CoV and SARS-CoV-2 M^pro^, data from MD simulations showed that the maximal accessible volume (MAV) of the binding cavities is significantly different. Indeed, the MAV value of SARS-CoV is over 50% wider than that of SARS-CoV-2. The accessibility of water molecules inside the enzymatic cavity was also analyzed, and it was possible to demonstrate how the amount of water that can enter the binding cavity is lower for SARS-CoV-2 M^pro^. This suggested the possibility of large conformational changes induced by the interaction with the ligand and solvent [22].

SARS-CoV-2 M^pro^ is a homodimer protease: Its structure is made up of two protomers (A and B) which, upon dimerization and activation, orient themselves in the appropriate conformation to perform the catalytic function [24]. Although most studies considered the presence of only two protomers, the presence of a third protomer (C) has been recently demonstrated, helping to clarify the stereochemistry towards the inhibition of SARS-CoV-2 M^pro^ by peptidyl aldehydes [25]. Similarly to many other cysteine proteases, each protomer is divided into 3 domains: The domains I (residues 8–101) and II (residues 102–184) consist of antiparallel β-barrels, and together they form the chymotrypsin-like structure [12,26,27,28,29]; the domain III (201–306), which is mostly composed by α-helices, is responsible for the catalytic process [26]. An interesting difference between the two types of structure is represented by the presence of a hydrogen bond between two residues of Thr285 of the two domains III (Figure 3) which, along with hydrophobic interactions between Thr285 and Ile286, keeps unified the two protomers of SARS-CoV M^pro^. In SARS-CoV-2 M^pro^, these two residues of Thr and Ile are replaced by Ala and Leu, respectively, and this maintains closer the two domains III [12,20], leading to a significant enhancement in the catalytic activity of the enzyme (3,6 fold), with a consequent increase in the catalytic turnover [30,31]. This might constitute an assumption for the greater contagiousness of SARS-CoV-2 compared to its ancestor.

The active site of the enzyme consists of 4 pockets (S1’, S1, S2, S3) [32], with the S1′ pocket containing a catalytic dyad [33]. This catalytic dyad is composed of the Cys145 and His41 residues and is inserted in a cleft between the domains I and II. The absence of the standard third catalytic element is compensated by the presence of a buried water molecule [18,33,34], which forms H-bonds with the residue of His41 and the surrounding amino acids. Another water molecule is located within the active site of the enzyme and establishes H-bonds with Phe140, His163, and Glu166, stabilizing the oxyanion hole [35]. Small-molecule inhibitors can replace the two water molecules inside the binding site, and this is probably associated with significant increase of potency [12,20,36,37,38]. The cleavage of viral polyproteins by M^pro^ involves the Leu-Gln(Ser, Ala, Gly) sequence, a characteristic motif shared among the coronaviruses. There are no human host cell proteases that perform this cleavage path. Therefore, it can be reasonably speculated that peptidomimetics and small-molecule inhibitors of M^pro^ might be extremely selective [13,14]. The M^pro^ substrate specificity requires the presence of a Gln residue at P1 [39], while at P2, large hydrophobic residues are generally preferred (e.g., aromatic rings capable of interacting with the target through π-π stacking interactions); on the other hand, the P1′ subsite usually tolerates small hydrophobic residues [17,40].

Cys145 and His41 form a catalytic dyad in which the thiol (-SH) group of cysteine is responsible for the hydrolysis [33,34]. His41 provides the optimal pH conditions necessary for the activation of the -SH group, which in turn realizes the nucleophilic attack to the substrate. The whole process carried out by the M^pro^ is depicted in Figure 4 and is divided in four phases: (1) Deprotonation of the -SH group of the Cys145 by His41, with formation of the activated thiolated ion; (2) nucleophilic attack of the latter to the carbonyl carbon of the substrate and formation of a tetrahedral adduct; (3) release of the peptide hydrolysis product with the free *N*-terminus and regeneration of the catalytic His41; (4) thioester hydrolysis with release of the remaining peptide fragment with the free *C*-terminus [41,42].

## 3. SARS-CoV-2 M^pro^ Inhibitors

Given the similarity of M^pro^ with other viral cysteine proteases, various pharmacophores have been re-proposed in the attempt to find specific anti-COVID-19 drugs. The discussion about the M^pro^ inhibitors can be divided into:FDA-approved drugs, proposed for the treatment of COVID-19 (Drug Repurposing).Peptidomimetic and non-peptidic compounds (small-molecules), derived from studies on SARS-CoV M^pro^ and other viral/retroviral targets [41].

### 3.1. Drug Repurposing

The development of a novel, selective, and safe drug for the treatment of COVID-19 is certainly the most coherent choice to combat the new epidemic. However, this is a long-term approach that does not fit with the current emergency. Therefore, the use of already available antiviral drugs was the most rational decision by the World Health Organization (WHO) in early 2020 as the first-line of intervention to deal with the impending pandemic. Due to their success in the treatment of previous coronavirus infections, the first idea was to re-propose existing antiviral drugs used in the treatment of HIV, hepatitis B and C, influenza, and the common cold [43]. The repurposing of FDA-approved drugs allows speeding up the experimental phases of a new therapy, since safety studies have already been validated [44]. However, the effectiveness of these drugs against COVID-19 has not been fully demonstrated to date. Clinical studies showed conflicting results, and the use of repurposed drugs for COVID-19 has often been futile, with no real impact on the patients’ prognosis. Another significant approach relies on the use of potential SARS-CoV-2 M^pro^ inhibitors (identified by virtual screening) which showed a polypharmacology profile in clinical trials for COVID-19 treatment that might be beneficial for patients with comorbidities [45].

The most important clinical trial carried out during the pandemic was SOLIDARITY, which sought to repurpose “old” antiviral molecules to ascertain their effectiveness against SARS-CoV-2 [46]. This trial proved that the lopinavir/ritonavir combination and the common antimalarial drug chloroquine were the best candidates in the treatment of early stage COVID-19. Unfortunately, further studies, including those ones carried out within the clinical trial RECOVERY, showed controversial results. Therefore, the therapeutic utility of these drugs, as well as that of many other repurposed drugs, is still under investigation [47,48]. Although it has often been proven to be a successful practice, the repurposing of drugs already approved for other therapeutic treatments is devoid of rationality. In fact, the lack of conclusive efficacy of these drugs often depends on their ability to affect other biological targets partially involved with the progress of the intended disease and/or the onset of symptoms. In the following section we provided a description of the most important drugs proposed for the treatment of SARS-CoV-2 infection that presumably target SARS-CoV-2 M^pro^. We have only briefly mentioned drugs inhibiting other viral targets, as it is beyond the scope of this review.

Table 1 summarizes the FDA-approved drugs repurposed for the treatment of COVID-19. Below, the main features concerning them.

**Lopinavir/ritonavir**. The lopinavir/ritonavir association is used as first and second line therapy in the treatment of human immunodeficiency virus (HIV) infections. These molecules belong to the HIV protease inhibitors class, and they reduce HIV infectivity and block the progress of the disease. Although ritonavir is itself considered a protease inhibitor, it is used in combination with lopinavir as it is also a cytochrome p450 3A4 inhibitor and therefore markedly increases the plasma concentration of other anti-HIV drugs [49]. Since ritonavir and lopinavir were active against SARS-CoV and MERS-CoV infections, they were initially proposed for the treatment of COVID-19. However, two studies published on May and June 2020, respectively, clarifies how the lopinavir/ritonavir combination was not effective in reducing mortality to 28 days in patients with COVID-19 [47,48]. Although they were prescribed ad hoc to reduce SARS-CoV-2 infection, these drugs are no longer used, and they require further clinical evidences to be officially approved. Numerous computational studies showed that these two drugs interact rather well with SARS-CoV-2 M^pro^ [50,51]. Unfortunately, their potency is still not sufficient to determine an adequate reduction of the infection. Yamamoto N. et al. suggested that the lack of efficacy of the lopinavir/ritonavir association relies in the insufficient ratio between Cmax/EC_50_ and proposed, based on in vitro studies, nelfinavir as an alternative [52]. Lopinavir and ritonavir have been designed as inhibitors of an aspartic protease (HIV protease), which has a completely different structure and mechanism of action compared to SARS-CoV-2 M^pro^. Furthermore, from a computational point of view, docking studies have shown that ritonavir possesses a low affinity score (−6.4 kcal/mol) in comparison to other HIV protease inhibitors docked on M^pro^. While the IC_50_ value of lopinavir proved to be rather unfavorable, ritonavir was the most effective among the panel of tested compounds by Mahdi M. and co-workers, with IC_50_ of 13.7 µM [53,54]. Therefore, lopinavir was progressively replaced with other commercially available antiviral drugs in the protease inhibitor combinations.

**Danoprevir**. It is a commercially available protease inhibitor for the treatment of hepatitis C virus infection. Based on the structural analogy between the HCV protease and SARS-CoV-2 M^pro^, a recently published study suggested danoprevir as a new potential anti-COVID-19 drug [55]. The results of the study, which involved 11 Chinese adult patients, demonstrated that the danoprevir/ritonavir combination reduce the symptomatology of the infection and accelerate the healing of COVID-19.

**Atazanavir**. In order to find a valid substitute for the lopinavir/ritonavir association, Fintelman-Rodrigues N. et al. proposed atazanavir as a new anti-COVID-19 drug due to its high potency, both alone and in association with ritonavir (EC_50_ = 2.0 µM and 0.5 µM, respectively) [56]. The atazanavir/ritonavir association has proven to be highly effective in inhibiting viral replication in Vero E6 cells, even better than chloroquine (EC_50_ = 1.0 µM). In addition, the same research team demonstrated that atazanavir is able to inhibit viral replication in human epithelial pulmonary cell lines (A549; EC_50_ = 0.6 µM) and to induce a reduction of IL-6 and TNF-α levels from monocytes. Therefore, the decrease of inflammatory processes translates into a prevention of cell death induced by SARS-CoV-2 infection. The mechanism of action of atazanavir relies in its ability to inhibit SARS-CoV-2 M^pro^. Computational studies have clarified how the interaction between this drug and M^pro^ does not occur at the catalytic site. The drug probably binds to the enzyme through several H-bond interactions [56].

**Nelfinavir**. Nelfinavir is a potent antiretroviral drug used to treat HIV infections in children and adults. Its clinical efficacy has not yet been proven. However, there are several in vitro studies showing the effectiveness of this drug in reducing SARS-CoV-2 replication [57]. Yamamoto N. et al. tested nelfinavir against Vero E6 cells infected with SARS-CoV-2, and the assay showed an EC_50_ of 1.13 µM, a better result compared to lopinavir (EC_50_ = 5.73 µM) [52]. Besides, numerous computational studies demonstrated that nelfinavir is a multi-target inhibitor, and this might validate its high potency in vitro [58,59].

**Boceprevir**. Boceprevir is a protease inhibitor approved by FDA in 2011 for the treatment of hepatitis C virus (HCV) [60,61,62]. Ma C. et al. showed that it potently inhibits SARS-CoV-2 M^pro^ (IC_50_ = 4.13 µM) by means of its reactive electophilic warhead (α-ketoamide moiety) and the resulting binding mode to the active site of the protease was confirmed by thermal shift binding assay (ΔTm = 6.67 °C), kinetic studies and native mass spectrometry [25]. The same research group determined that boceprevir is able to markedly reduce the replication of SARS-CoV-2 (EC_50_ = 1.90 µM) by cellular viral cytophatic effect (CPE) assay. Therefore, this M^pro^ inhibitor would be worth considering for further evaluation in clinical trials and in vivo studies.

**Carmofur**. This derivative of 5-fuorouracil (5-FU) bearing a 1-hexylcarbamoylic chain, inhibits SARS-CoV-2 M^pro^ with an IC_50_ of 1.82 µM. To date, it is believed that it acts as an antineoplastic drug via inhibition of the thymidylate synthetase, a key enzyme involved in cell replication processes. However, it has been observed that carmofur is able to inhibit human acid ceramidase by acylation of its cysteine catalytic residue [38]. Besides, carmofur was shown to be effective in a cell-based antiviral assay with an EC_50_ of 24.87 μM [38]. X-ray analysis showed how carmofur interacts with SARS-CoV-2 M^pro^ by covalent modification of the residue of Cys145, acylating the thiol group and releasing the 5-FU (PDB 7buy). The carbonyl oxygen interacts with Gly143 and Cys145 by H-bond, and the formed adduct mimics the transition state of the enzyme-substrate complex. The alkyl chain of six carbon atoms is stabilized by hydrophobic interactions with several amino acids of the enzyme pocket and it inserts in the S2 subsite cavity. Unfortunately, being a small and flexible molecule, carmofur is unable to interact deeply within the active site. Nevertheless, carmofur can be taken into account for subsequent structural modifications for the development of potential new SARS-CoV-2 M^pro^ inhibitors [20].

**Ebselen**. The best candidate identified so far for the treatment of COVID-19 (IC_50_ = 0.67 µM) appears to be ebselen, a synthetic organoselenium compound with anti-inflammatory, antioxidant, and cytoprotective activities [63,64,65]. It acts as a promiscuous binder. Although its covalent inhibition mechanism of the SARS-CoV-2 M^pro^ has been demonstrated, this compound can interact with the target also in a non-covalent fashion, a fact that would explain the high IC_50_ observed [20]. Moreover, ebselen was demonstrated to reduce CPE of Vero E6 cells infected with SARS-CoV-2 (EC_50_ = 4.67 µM) [66].

**Remdesivir**. During the initial stage of the pandemic, remdesivir was considered the best candidate for the treatment of COVID-19 [67,68]. However, in November 2020, the WHO cautioned about the indiscriminate use of this drug as additional studies did not show substantial improvements in the life expectancy of patients affected by COVID-19. More recent results, obtained from comparative effectiveness research studies of adults hospitalized with COVID-19, instead indicated that remdesivir was associated with faster clinical improvement [69]. This molecule potently inhibits the replication of all types of coronaviruses in vitro, and it is therefore considered the reference drug for the development of new anti-coronavirus agents. It was originally used for the treatment of the Ebola virus, and it has been proven to be active against other viral strains, in particular retroviruses. Remdesivir is the first and only FDA-approved treatment for COVID-19 in the US [70], and it has been approved or authorized for temporary use as a COVID-19 treatment in approximately 50 countries worldwide [71]. The mechanism of action of remdesivir entails its conversion into an active metabolite (GS-441524; structural analogue of the adenosine) which blocks viral replication by inhibiting the viral RNA-polymerase-RNA-dependent, probably acting as a chain terminator or causing mutations in the genoma of the virus. In vitro remdesivir inhibits the replication of SARS-CoV-2 with an EC_50_ of 0.77 µM and CC_50_ > 100 in Vero E6 cells (SI = 129.87) [72], and has proven to be effective in a few severely ill patients [67]. Computational studies highlighted how remdesivir and its metabolite can efficaciously form complexes with SARS-CoV-2 M^pro^ showing affinity scores of −7.0 and −6.4 kcal/mol, respectively [73].

**Dipyridamole**. Dipyridamole is a coronary vasodilator drug, which has been used clinically as an effective anti-COVID-19 drug [74]. It possess a low EC_50_ value (0.1 µM) against SARS-CoV-2 in vitro, and the mechanism of action involves the inhibition of SARS-CoV-2 M^pro^. However, computational studies have not demonstrated a good interaction between M^pro^ and dipyridamole (−5.4 kcal/mol) [74], probably due to the presence of a DMSO molecule that occupies the dipyridamole binding site, translating the result into a false negative [73]. Further studies are needed to investigate the mechanism of action because of its promising EC_50_ value.

**Chloroquine/Hydroxychloroquine**. Chloroquine has been shown to inhibit SARS-CoV-2 in vitro (EC_50_ = 5.4 µM) [72,75,76,77,78]. Chloroquine and hydroxychloroquine act as potent inhibitors of SARS-CoV-2 M^pro^, with K_i_ of 0.56 μM and 0.36 μM, respectively [79]. Docking and dynamics simulation studies indicated also that both compounds affect other viral targets [80], and therefore they have been proposed for the treatment of COVID-19. However, there is still no evidence for their efficacy in vivo.

### 3.2. Peptidomimetic Inhibitors

Peptidomimetics are synthetic tools used to mimic the natural structure of a peptide or protein and interact with a biological target the same way as the original substrate. They show significant improved characteristics compared to natural peptides, including metabolism stability, better pharmacokinetic properties, stronger potency, etc. [81]. To date, the development of peptidomimetics is the most used strategy for the search of anti-COVID-19 drugs [82,83]. Indeed, several SARS-CoV M^pro^ inhibitors developed over the past decade are peptidomimetics, and then their re-proposition as SARS-CoV-2 M^pro^ inhibitors has been a logic strategy, also in the view of the above-referred similarity between the two targets [41].

SAR studies carried out on small peptide-based cysteine proteases inhibitors demonstrated that all of them share similar features. The general inhibitor structure can be divided into 4 portions (P1’, P1, P2, P3). P1-P3 constitutes the peptide backbone, which may contain modified amino acids to increase peptidomimetic characteristic, and provides the specific recognition motif for SARS-CoV-2 M^pro^. A *C*-terminal electrophilic function (warhead) occupies the P1′ site, and it is required for the covalent inactivation of the protease. These *C*-terminal electrophilic warheads can be classified according to the different functional groups present at P1′ into [41]:Aldehydes;α-Ketoamides;Vinyl esters;Fluoromethyl ketones (FMKs);Hydroxymethyl ketones;Acyloxymethyl ketones.

Inhibition occurs via a two-stage mechanism: i) A reversible first stage which entails an association between enzyme and inhibitor, involving interactions between the amino acids of the binding pocket and those of the peptide backbone; ii) a second stage represented by the nucleophilic attack of the thiol group to the most activated carbon atom of the inhibitor leading to the formation of a new bond. Depending on the strength of the newly formed covalent bond, the inhibition can be reversible, if a water molecule present in the enzyme cavity can free the enzyme from its inhibitor restoring the protease activity, or irreversible if it remains permanently bound to the protein. Generally, an irreversible mechanism for peptidomimetics bearing Michael acceptor warheads and a reversible mechanism for peptidyl aldehydes has been observed. An exhaustive description of the structure, inhibition mechanism, and main biological data in vitro of the peptidomimetics proposed as potential anti-COVID-19 drugs is summarized in the next section.

#### 3.2.1. Peptidyl Aldehydes

The use of peptidyl aldehydes as viral cysteine protease inhibitors (including SARS-CoV M^pro^) is associated to the high reactivity of the aldehyde group [84,85], which undergoes nucleophilic attack of the activated cysteine thiolate and forms a reversible covalent hemi-thioacetal adduct [86]. Their superior activity with respect to Michael acceptors is ascribed to the maintenance of the carbonyl group at P1, which plays a fundamental role as an acceptor of H-bond within the binding pocket. Furthermore, the selectivity of peptidyl aldehydes towards exogenous cysteine proteases is marked. Therefore, these compounds can be envisaged as excellent anti-COVID-19 agents.

Dai W. et al. discovered two potent SARS-CoV-2 M^pro^ inhibitors bearing an aldehydic warhead (**1** and **2**, Figure 5a–c) [37]. These compounds exhibit at the P1 site a γ-lactam moiety, as commonly observed in other SARS-CoV-2 M^pro^ inhibitors employed as a bioisostere of Gln, and at P2 a hydrophobic amino acid with a cyclohexyl or aryl as side chain. The P2-P3 peptide bond was obtained by joining the amino group of the P2 amino acid with 2-indol-carboxylic acid, chosen for the possibility of providing further interactions with the target by means of an additional H-bond and improving the drug-like properties of these peptidomimetics. **1** and **2** showed powerful inhibition against SARS-CoV-2 M^pro^ with IC_50_ of 53 nM and 40 nM, respectively. Treatment of Vero E6 cells infected with SARS-CoV-2 with the two inhibitors **1** and **2** evidenced a significant decrease of the viral replication capacity expressed as EC_50_ values of 0.42 μM and 0.33 μM, respectively. The crystallographic structure of the two inhibitors with SARS-CoV-2 M^pro^ (PDB **1** 6lze, PDB **2** 6m0k) showed (as expected) the presence of a covalent bond between the aldehyde carbonyl and -SH group of Cys145. Furthermore, the resulting thiohemiacetal function is stabilized by the presence of several H-bonds between the aldehyde oxygen and Cys145 and Gly143 of S1′ subsite. The adduct is also stabilized by various water molecules, mimicking the enzyme-substrate transition state. Similarly to other inhibitors, the γ-lactam moiety in P1 interacts with several residues (i.e., Phe, His and Glu) of the S1 subsite, establishing multiple H-bonds. Moreover, the hydrophobic side chain of the P2 site (cyclohexyl for compound **1** and aryl of compound **2**) forms bonds with aliphatic amino acids of the target through van der Waals and π-π stacking interactions. Finally, the indole group interacts with a residue of Glu166 and faces the S4 subsite where it is exposed to the solvent.

Compound **3** (Figure 6a) is a veterinary drug used for the treatment of the feline infectious peritonitis (FIP) in cats [87]. FIP can be often fatal to cats and is caused by the feline coronavirus FCoV. **3** is a bisulfite aldehyde inhibitor of FCoV-M^pro^ which has been re-proposed in the treatment of COVID-19 in view of its successful outcomes in animals [88,89]. The mechanism of action of the inhibitor involved its conversion into the active derivative **4** (peptidyl aldehyde), which in turn forms the classic covalent adduct with the -SH group of Cys145. IC_50_ measurements demonstrated that **3** is able to inhibit SARS-CoV-2 M^pro^ in the low-micromolar range (IC_50_ = 0.03 μM) [72]. The aldehyde derivative **4** turned out to also be capable to inhibit M^pro^ though to a lesser extent (IC_50_ = 0.40 μM) [90]. However, the use of the bisulfite form would be preferred due to better perspectives in terms of bioavailability and toxicity profile. X-ray studies have clarified the most important interactions between **4** and the M^pro^ catalytic site (Figure 6b): The inhibitor acts by forming an intense network of H-bonds with the biological target. The P1 site contains a γ-lactam moiety (as described before), which interacts via H-bonds with the side chains of His163 and Glu166 and with the main chain of Phe140. The P2 site is characterized by the presence of a Leu residue, which enters its isobutyl group into the S2 subsite establishing hydrophobic interactions with His41, Met49, and Met169. The carbamate moiety forms H-bonds with the main chain of Glu166 and with the side chain of Gln189. The benzyl group at the *N*-terminus is bound to the S4 subsite through extensive hydrophobic interactions, and therefore the presence of this functional group is responsible for the high potency to the inhibitor. The X-ray structure of SARS-CoV-2 M^pro^ with **4** has shown that the compound can orient itself in the enzymatic pocket by means of two different P1 configurations, R or S, according to the nucleophilic attack of the thiol residue of Cys145 to the planar carbonyl of the aldehyde. In fact, three crystallographic structures have been highlighted, in which the inhibitor binds to the Cys145 residue, one for each protein protomer. For protomer A and B, the interaction enzyme-inhibitor occurred through the “face R”, unlike the protomer C, which formed a more stable covalent adduct with S stereochemistry at P1 [72].

Yang K.S. et al. took **4** as lead compound for the development of reversible tripeptide inhibitors bearing an aldehyde structure [91]. **5** (Figure 7a,b in complex with the target) provided the best results in terms of enzymatic inhibition (IC_50_ = 8.5 nM), while **6** and **7** (Figure 8) proved to be the best substrates capable of inhibiting SARS-CoV-2 replication in vitro on Vero E6 cells, with higher EC_50s_ than the reference compound **3**. This outstanding work of Yang’s research group may represent an excellent starting point for the further development and optimization of anti-coronavirus peptidomimetics.

Among aldehyde-based peptidomimetics, numerous calpain inhibitors have been developed to date [92]. Ma C. et al. evaluated the activity of various calpain inhibitors for repurposing them as anti-COVID-19 drugs. The most active compound is represented by **8** (Figure 9, IC_50_ = 0.97 μM) whose structure of consists in two lipophilic substituents (Leu) at P2 and P3. Unlike other aldehyde inhibitors, this compound is characterized by the presence of an acetyl group (instead of Cbz) at the *N*-terminus. However, this modification seemed to decrease the inhibitory activity against M^pro^, in comparison with other calpain inhibitors. Evaluation of this peptidomimetic against infected Vero E6 cells showed a good antiviral profile (EC_50_ = 2.07 μM) [25].

#### 3.2.2. α-Ketoamides

Zhang L. et al. showed that peptidomimetics with an α-ketoamide recognition motif manifested an inhibitory activity against SARS-CoV-2 M^pro^ [12,13]. They reported the X-ray structure of both the unliganded protease and the protease bound to the α-ketoamide inhibitor, highlighting the main interactions within the enzymatic cavity.

Compound **9** (Scheme 1) was originally designed and synthesized as a MERS and SARS-CoV proteases inhibitor with IC_50_ values in the picomolar and micromolar range, respectively [13]. Starting from the lead compound **9**, Zhang L. and co-workers made several changes to afford **10**. In the latter, in order to increase the plasma stability and reduce the catabolic processes, the P2 and P3 sites were constrained into a pyridone ring. The cinnamoyl linker of **9** was replaced in **10** with the less bulky Boc group to decrease the binding affinity of the molecule with plasma proteins. In addition, the P2 cyclohexyl side chain, originally designed for selectivity requirements towards enterovirus, was replaced by smaller groups such as cyclopropyl to afford **11** and enhance the inhibitory activity.

The crystal structure of **11** bound to SARS-CoV-2 M^pro^ (Figure 10) showed how the inhibition occurs through the formation of a covalent interaction between Cys145 and the α-keto group of the inhibitor (PDB 6y2f). The formation of the thioemiketal adduct consists in a reversible event, but compared to other inhibitors such as Michael’s acceptors, the possibility of the amide group to establish H-bonds increases the interaction with the target, and the resulting inhibition is much stronger. As for the majority of peptidyl aldehydes, the side chain of the P1 site is occupied by a γ-lactam moiety, which interacts with the biological target through several H-bonds. Specifically, the lactam nitrogen of the Gln surrogate donates a three-center H-bond (bifurcated) to Phe140, Glu166, and His163 (Figure 10). As previously observed, the preferred amino acid at the P1 site is Gln. However, several α-substituted ketones undergo cyclization and inactivation. Replacing the Gln residue with a lactam moiety solves the problem, avoiding the unwanted cyclization, and not altering the binding with the protease. P2 cyclohexyl or cyclopropyl side chains interact with hydrophobic amino acids of the pocket and the 14yridine oxygen accepts a hydrogen bond from His166. Finally, the elimination of the Boc group in **12** resulted in a total loss of activity. **11** blocked the protease activity of SARS-CoV-2 M^pro^ with an IC_50_ of 0.67 μM and was able to inhibit the proliferation of the virus in human Calu3 lung cells, unlike **12**, which proved to be completely inactive.

As above-mentioned, Ma’s research group suggested calpain inhibitors as rich sources of drug candidates for SARS-CoV-2 [25]. The best α-ketoamide derivative identified as a potential anti-COVID-19 agent was **13** (Figure 11). This compound showed excellent antiviral activity in the cell-based assay (EC_50_ = 0.49 μM in Vero E6 cells) and inhibitory activity against M^pro^ (IC_50_ = 0.45 μM). The remarkable efficacy of these calpain inhibitors to block the enzyme activity could be associated with their ability of acting as dual-target molecules. Calpain II and cathepsin L are fundamental enzymes for the processing of the coronavirus S protein, a necessary step for the virus to penetrate the host cell [93]. Therefore, the simultaneous inhibition of calpain and a cathepsin-like enzyme such as M^pro^ may result in an increase of the antiviral activity and reduce the development of resistance. For this reason, repurposing of calpain/cathepsin inhibitors for SARS-CoV-2 infections should be implemented.

#### 3.2.3. Vinyl Esters

Within this class of derivatives, Jin Z. and co-workers identified compound **14** (Figure 12a) as a lead structure for the development of novel M^pro^ inhibitors [20]. Inhibitor **14** (named N3) is a pseudo-tetrapeptide, modified in P1 with a α,β-unsaturated benzyl ester, discovered via computer-based drug design [94]. It belongs to the Michael’s acceptors, a class of serine and cysteine protease inhibitors that react by forming an irreversible covalent bond with the enzyme thanks to the presence of the activated double bond (Figure 12b) [95].

**14** was originally designed as inhibitor of MERS-CoV and SARS-CoV proteases and showed potent antiviral activity in animal models of bronchitis. Molecular docking studies predicted how this M^pro^ inhibitor could fit within the enzymatic cavity. The mechanism of action of **14** consists of an irreversible two-step inhibition. Initially, the inhibitor associates with the protease in a reversible way; subsequently, the attack of the activated thiol group occurs at the β-C of the conjugated double bond, leading to the formation of an irreversible covalent bond. To assess the efficacy of **14** towards SARS-CoV-2 M^pro^, Jin’s group performed several kinetic analysis and demonstrated how the molecule act as a time-dependent irreversible inhibitor with *k*_2nd_ of 11,300 M^−1^ s^−1^ [20]. The antiviral activity of **14** in the cell-based assay led to an EC_50_ of 16.77 μM.

#### 3.2.4. Peptidyl Fluoromethyl Ketones (pFMKs)

Peptidyl fluoromethyl ketones (pFMKs) have always been substrates capable of inhibiting viral and parasitic cysteine proteases [96,97], including the M^pro^ of the coronavirus family. The presence of fluorine atoms in the *C*-terminal α position of a methyl ketone moiety enhances the susceptibility of carbonyl group to undergo a nucleophilic attack by the protease thiol group. On the basis of the number of the fluorine atoms in this position we can distinguish three different classes of compound namely mono-fluoromethyl ketones (MFMKs), di-fluoromethyl ketones (DFMKs), and tri-fluoromethyl ketones (TFMKs).

Recently, Zhu W. and collaborators identified the MFMK **15** (Figure 13a), an irreversible inhibitor (SN2 displacement of the fluorine atom) of caspase-3 as a potent inhibitor of SARS-CoV-2 M^pro^ (IC_50_ = 11.39 μM). **15** also showed a marked antiviral profile in the CPE assay against SARS-CoV-2 infected Vero E6 cells (EC_50_ = 0.13 μM) [98]. The predicted binding mode of **15** inside M^pro^ pocket is depicted in Figure 13b. However, the therapeutic utility of MFMKs is compromised due to drawbacks related to their metabolism (formation of toxic fluoroacetate). Therefore, their use is essential limited to pharmacological tools as activity-based probes with selectivity toward cysteine proteases [96].

DFMKs and TFMKS instead appear to be much more promising with respect to MFMKs as therapeutics in view of their ability to target also serine proteases (increased electrophilicity of the carbonyl group) with no significant metabolic issues. Both fluorinated-type warheads may hit the target by a slow-binding competitive reversible mechanism or act as transition-state competitive analogues because they may stand in the hydrated form [96]. In this regard, the first notable outcome was obtained in 2008 by Shao Y.-M. and co-workers which developed a series of dipeptidyl TFMKs as good inhibitors of SARS-CoV M^pro^ in vitro [99]. The most active derivative **16**, drawn in the predominant hydrated form, is depicted in Figure 14 and showed activity in the micromolar range (IC_50_ = 10 µM).

On this premise, our research group recently applied the chemistry of α-fluorinated organometallic methyl type-carbanions [100,101], to synthesize the dipeptidyl DFMK **17**, structurally related to **16**, having the sequence Z-Leu-Homophe- as peptidic framework connected to the *C*-terminal -COCHF_2_ moiety. Compound **17** showed a remarkable antiviral activity (EC50 = 12.9 μM) on cells infected with hCoV-229E, one of the four human coronaviruses associated with respiratory distress. Molecular docking and molecular dynamics studies indicated that **17** might efficaciously bind to the intended cysteine target of both hCoV-229E and SARS-CoV-2. This DFMK represents the reference structure of our ongoing research aimed at the design of druggable compounds with activity against coronaviruses [97].

#### 3.2.5. Hydroxymethyl Ketones

A series of ketone derivatives targeting SARS-CoV-2 M^pro^ was recently investigated by the Pfizer Inc. [102]. From this study, hydroxymethyl ketones emerged as potent M^pro^ inhibitors. Specifically, **19** (PF-00835231; Figure 15a) was selected for further development as COVID-19 treatment for its potent inhibition of the M^pro^ and viral replication of SARS-CoV-2 and other coronaviruses. The favorable pharmacokinetic parameters of **19**, including solubility and metabolic stability, are consistent with the intravenous administration. Currently, also **18** (PF-07304814; Figure 15a), the phosphate prodrug of **19**, is under phase I clinical trials [103]. **19** was developed in 2003 during SARS emergency, but its clinical advancement was suspended at the end of the pandemic and it was reproposed again following COVID-19 outbreak. Biological evaluations confirmed that **19** blocked the SARS CoV-2 M^pro^ activity (K*_i_* = 0.27 nM) as evidenced by the cocrystal structure of the inhibitor covalently bound in the active site of the protease (Figure 15b). Vero E6 kidney cells enriched for angiotensin-converting enzyme 2 (ACE2) receptor (Vero E6-enACE2) and Vero E6 cells constitutively expressing EGFP (Vero E6-EGFP) were used for the evaluation. CPE assay demonstrated how the inhibitory activity was more than 100 times higher when **19** was co-administrated with a P-gp efflux inhibitor [EC_50_ = 0.23 µM (Vero E6–enACE2 cells); EC_50_ = 0.76 µM (Vero E6-EGFP cells)], suggesting that the infected cell may implement drug efflux as a resistance mechanism.

#### 3.2.6. Acyloxymethyl Ketones

Peptidomimetic-based probes are useful tools for studying substrate activity and specificity as well as for the detection of a protein inside a biological sample [104,105,106]. Activity-based probes (ABPs) show some unique characteristics: (i) The ability to form a stable, often covalent, bond with the substrate, achieved by introducing an electrophilic *C*-terminal warhead to the peptide chain; (ii) the presence of a detectable group, generally a fluorophore; (iii) the substrate specificity, given by the selective interaction between the peptide backbone and the biological target. ABPs bearing an acyloxymethyl ketone moiety were the first probes synthesized for the detection of SARS-CoV-2 M^pro^ [107]. Besides the warhead, they are characterized by the presence of a peptidomimetic backbone bearing amino acid residues for the specific recognition of SARS-CoV-2 M^pro^. Probe **20** (Figure 16) was demonstrated to be the most potent, bearing a Gln residue at P1, modified by replacing the two H with methyl groups, to reduce cyclization and inactivation phenomena [41]. M^pro^ was found covalently inhibited by probe **20** at concentrations lower than 200 nM. Meanwhile, 9% of residual activity was recorded after 1 h of incubation with probe **20** at 10 μM concentration—with carmofur as a control—demonstrating that acyloxymethyl ketones could potentially be excellent covalent active site inhibitors of SARS-CoV-2 viral replication. “Click” chemistry (copper-catalyzed azide-alkyne cycloaddition) was used to connect the terminal alkynyl portion of the inhibitor to a fluorophore in order to evaluate the covalent labeling of SARS-CoV-2 M^pro^, and the results of Merel van de Plassche’s work have demonstrated a strong selectivity of the acyloxymethyl ketone-based probes [107].

#### 3.2.7. Cyclic Peptides

A first-in-class cyclic peptide targeting SARS-CoV-2 M^pro^ was recently proposed by Nowick J.S. et al. [108]. The hit compound called **UCI-1** (Figure 17) was designed to mimic the conformation of linear peptide substrates of M^pro^ at its *C*-terminal autolytic cleavage site. Specifically, **UCI-1** contains the amino acid side chains inferred from the P2, P1, P1′, and P2′ positions of the M^pro^ substrate and designed to fill the four enzymatic pockets. Moreover, a [4-(2-aminoethyl)phenyl]-acetic acid (AEPA) group, designed to act as a surrogate for the Phe side chain, connects the *C*-terminus of the P2′ residue to the *N*-terminus of the P2 residue creating a cyclophane. **UCI-1** was synthesized by Fmoc-based solid-phase peptide synthesis and purified using reverse-phase HPLC. A traditional lactase dehydrogenase assay excluded cytotoxicity on human embryonic kidney (HEK-293) cells at concentrations up to 256 μM. The enzymatic inhibition was evaluated using a fluorescence-based M^pro^ inhibition assay kit (BPS Bioscience) that included purified SARS-CoV-2 M^pro^ as a fusion protein with maltose binding protein (MBP-M^pro^) and a fluorogenic M^pro^ substrate. Specifically, the activity of MBP-M^pro^ was measured in the presence of increasing concentrations of **UCI-1**, revealing an IC_50_ of ~150 μM, and was compared with those of a linear control analogue, that showed little or no inhibition. These results confirmed that the cyclic structure of **UCI-1** is crucial for M^pro^ inhibition. Despite the activity of **UCI-1** is modest compared to other known inhibitors, the study of Nowick et al. could open the door to the rational design of additional cyclic peptide inhibitors analogs of **UCI-1** with improved activity against M^pro^.

### 3.3. Non-Peptidic Inhibitors

As can be deduced from the repurposing approach, also non-peptidic inhibitors have been recently proposed as potential anti-COVID-19 drugs [20]. Recent studies of molecular dynamics coupled with protein stability analyses have elucidated that there are substantial differences in the plasticity and accessibility of the active site SARS-CoV-2 M^pro^ in comparison with that of SARS-CoV M^pro^. The obtained results evidenced that small-molecule inhibitors might be preferred as lead compounds for the design of anti-COVID-19 drugs [22]. According to the crystallographic data deposited on PDB, most of the proposed small-molecule inhibitors of M^pro^ bear an electrophilic reactive function (ketone, amide, vinyl ester, inter alia) within their structure that might bind and inactivate the active site Cys residue of M^pro^. However, their biological evaluation is still ongoing in most cases. To date, as far as we know, compounds with a tetralonic, quinonic, coumarinic, and heterocyclic template could behave as valid SARS-CoV-2 M^pro^ inhibitors. These structures are discussed below together with their features and mechanism of action.

A recent and extensive crystallographic study recruited 5632 individual compounds to evaluate their binding to the M^pro^ [109]. Among these screened compounds, it emerged that **21** (HEAT) could be a potential new anti-COVID-19 candidate (Figure 18a). The research team reported the X-ray structure of **22**, a tetralone derivative obtained by decomposition of **21**, in complex with M^pro^ (Figure 18b) [110].

In particular, this study showed how this derivative undergoes a metabolic conversion into a Michael acceptor derivative, which in turn interacts covalently with M^pro^’s Cys145 [110]. Within the binding pocket, His163 and Cys145 stabilize **21** in its enolic form. Subsequently, a molecular rearrangement (E1cB-like reaction mechanism; Figure 19) generates an α-β-unsaturated compound. The obtained 2-methylene-tetralone (**22**) finally undergoes a nucleophilic attack on the activated β-carbon with formation of a new covalent bond that results in an irreversible inhibition of the protease. This unique mechanism of action suggested that **21** could be used as a potential suicide inhibitor of SARS-CoV-2 M^pro^. Molecular modeling studies elucidated why the enolic tautomer of **21** appear to be more stable than the ketone tautomer within the catalytic site. This is because **21** forms a network of H-bonds with the surrounding amino acid residues [110].

Zhu W. et al. evaluated the antiviral activity of several small molecules towards Vero E6 cells infected with SARS-CoV-2, and the results showed that the antibacterial drug walrycin B (**23**, Figure 20) was able to reduce CPE with EC_50_ of 3.55 μM [98]. Structurally, **23** is endowed with a pyrimidotriazine-dione core and a *p*-trifluoromethylphenyl substituent. The M^pro^ assay in vitro showed a noteworthy inhibition of protease activity with an IC_50_ value of 0.26 μM.

The great importance of small molecules as M^pro^ inhibitors was also highlighted by the work published by Hattori S.-I. and collaborators in July 2020 [111]. The authors identified **24** (Figure 21) as a new potent inhibitor of SARS-CoV-2 M^pro^ able to block coronavirus infection. **24** is an indole chloropyridinyl ester which binds covalently to M^pro^ as underlined by the X-ray crystal structure of the complex enzyme-inhibitor reported by the same group. This heterocyclic derivative exerted potent antiviral activity (EC_50_ = 2.8 μM) towards infected Vero E6 cells without any significant cytotoxicity. The mechanism of action of **24**, confirmed via HPLC/MS studies, relied on the nucleophilic attack of the Cys145 thiol group to the ester carbonyl with the expulsion of the leaving group. The covalent binding of the inhibitor to the target was then supported by differential scanning fluorimetry experiments; the interaction of **24** with the protease produced a destabilization effect that shifted the T*_m_* of the enzyme to lower temperatures.

Chinese natural medicine offers a source of plant-derived molecules with antiviral properties, recently proposed as anti-COVID-19 remedies [112,113]. Shikonin (**25**, Figure 22a) is the major component extracted from the roots of *Lithospermum erythrorhizon* [114], a Chinese herb, and previous data have shown good inhibition activity towards SARS-CoV-2 M^pro^ (IC_50_ = 15.75 μM) [20]. **25** is a naphthoquinone derivative and it binds non-covalently to the M^pro^ active site. X-ray data of the shikonin-M^pro^ complex (PDB 7ca8) revealed that the nucleophilic attack of the activated thiol group of Cys145 to the compound does not occur. Instead, **25** inhibits M^pro^ through the formation of several H-bonds within the catalytic site (Figure 22b). The side chain of Cys145 adopts a different conformation, interacting with **25** trough H-bond. In addition, the imidazole group of His41 was found oriented in the opposite way in the catalytic site with respect other crystalline structures allowing the entry of the inhibitor. The reversible bond between the inhibitor and the protease therefore results in an unprecedented conformational modification of the catalytic dyad. Furthermore, **25** was able to bind only the protomer A, unlike all the other inhibitors described above. The most important interactions between shikonin and M^pro^ are shown in Figure 22. Li J. and coworkers debated that the inhibition of the enzyme by small molecules could proceed with a different mechanism [36]. Unfortunately, reversible inhibition of M^pro^ by **25** resulted in a low EC_50_ value (>100 μM); the antiviral effect of **25** against SARS-CoV-2 in Vero E6 cell culture was evaluated by Gurard-Levin and collaborators [115].

The roots of *Scutellaria baicalensis* have long been used in traditional Chinese medicine for the prophylaxis and treatment of numerous viral infections [116,117,118]. Two research groups evaluated the antiviral activity of *S. baicalensis* extracts and the main active ingredients, baicalin (**26**) and baicalein (**27**) (Figure 23), against Vero E6 cells infected with SARS-CoV-2 [119,120]. The results showed that the ethanol extract of the plant roots possess antiviral properties against SARS-CoV-2 (EC_50_ = 0.74 μg/mL), while **26** and **27** significantly reduce the growth of the virus (EC_50_ = 10.25 μM and EC_50_ = 1.69 μM, respectively) [120]. The mechanism of action of these flavonoids is based on their ability to inhibit non-covalently M^pro^. The IC_50_ value of **27** attested a remarkable inhibitory capacity of this flavonoid towards SARS-CoV-2 M^pro^ (IC_50_ = 0.39 μM) [119]. **26** showed a lower activity (IC_50_ = 6.41 μM) compared to **27**, and this is probably due to the presence of the sugar residue which did not fit well into the active site of the enzyme [120]. The X-ray structure of **27** with M^pro^ (PDB 6m2n) demonstrated how the flavonoid binds in an entirely different way compared to peptidomimetics: **27** enters deeply the binding pocket establishing several H-bonds and π-π stacking interactions with the catalytic dyad, the oxyanionic cavity, and the S1/S2 sites. By stabilizing the conformation of the oxyanion loop, it acts as a “shield” and prevents the approach of the inhibitor to the active site [120]. Four other flavonoids, analogues of **27**, have been tested as M^pro^ inhibitors. Among them, the most active derivative turned out to be scutellarein (**28**, Figure 23) (IC_50_ = 5.80 μM), a compound already known for being a replicase inhibitor [121]. Then, flavonoids, apart for their antioxidant properties, could be used as lead compounds in the design of new anti-COVID-19 drugs due to their high activity as M^pro^ inhibitors and low toxicity [122]. All compounds with potential anti-COVID-19 activity are depicted in Table 2.

## 4. Enzymatic Assays

For the assessment of the activity of M^pro^ inhibitors, some fluorimetric assay kits have been developed and are now commercially available. Pitsillou E. et al. employed a kit developed by the BPS Bioscience for an enzyme-linked immunosorbent assay (ELISA) [123], that includes as positive control the broad-spectrum antiviral GC-376 (IC_50_ = 0.46 μg/mL). Boras B. et al. exploited the biochemical Förster Resonance Energy Transfer (FRET) protease activity assay (i.e., SensoLyte^®^ 520 SARS-CoV-2 3CL Protease) for the evaluation of the potent SARS-CoV-2 inhibitor PF-00835231 [124]. In this test, the peptide substrate binds a fluorescent group and a quencher group to both ends of the sequence containing the cleavage site. When the active M^pro^ cleaves the substrate, FRET phenomenon results in the energy transfer by the two fluorophores and in an increase of 488 nm green fluorescence monitored at excitation/emission = 490 nm/520 nm. This assay can detect as low as 15.6 ng/mL active M^pro^ protease in the sample.

Issues related to the inhibition of cysteine proteases are the evaluation of the mode of action (specific or nonspecific) and the potential antiviral activity at cellular level. Standard protocols entail the addition of reducing reagent such as dithiothreitol (DTT), glutathione (GSH), or β-mercaptoethanol (β-ME) in the assay medium to avoid oxidation of the thiol groups with formation of disulfide bridges. Without the presence of the reducing reagent, an apparent inhibition might occur due to either such spontaneous oxidation or alkylation of the cysteine residue by reactive compounds. A series of techniques including FRET-based enzymatic assay, thermal shift assay and native mass spectrometry have been used by Ma C. et al. to investigate the specific or nonspecific inhibition of ebselen, disulfiram, tideglusib, carmofur, shikonin, and PX-12 against SARS-CoV-2 M^pro^ [125]. The authors demonstrated that the inhibition of M^pro^ by these six compounds is nonspecific, and that the inhibition is reduced or abolished with the addition of DTT, whereas without DTT they inhibited a panel of viral cysteine proteases including M^pro^, SARS-CoV-2 papain-like protease, and 2A^pro^ and 3C^pro^ from enterovirus A71 (EV-A71) and EV-D68.

The cloning, expression, and purification procedures for SARS-CoV-2 M^pro^ and SARS-CoV-2 PL^pro^ and the synthesis of FRET-based peptide substrates were described in the literature [25,96,125]. The procedure, in the presence or not of the reducing agent, requires the incubation of the protease with various concentrations of inhibitors at 30 °C for 30 min in the assay buffer and the subsequent addition of the FRET-based substrate to initiate the reaction. The reaction was monitored for 2 h, and the initial velocity was calculated using the data from the first 15 min by linear regression. The IC_50_ values were calculated by plotting the initial velocity against various concentrations of testing inhibitor.

To gain insights into the mechanism of action of the inhibitors and to investigate the specificity of the protease inhibition, the fluorimetric assays were coupled to thermal shift binding assay (TSA) and mass spectrometry (MS) binding assay. Both protocols should be carried out in the presence and absence of a reducing agent. In the TSA, a temperature gradient is applied to induce the denaturation of a protein. The experiment is carried out in the presence of a fluorescence dye that, by interacting with the exposed hydrophobic region of the unfolded protein, increases the fluorescent signal. The specific binding of an inhibitor to the native state of a protease usually stabilizes the protein leading to a shift of its melting temperature (ΔTm) [126]. MS-binding assay studies the formation of adducts between protease (i.e., monomeric or dimeric forms) and inhibitor. The results obtained from native mass spectra should be correlated with the mechanism of action unveiled by the X-ray crystallography.

To provide physiologically meaningful testing of molecules without the use of live SARS-CoV-2 virus at biosafety level 3 (BSL3), specific assays namely “surrogate assays” were investigated [127]. Resnick S.J. et al. reported that the expression of the M^pro^ in HEK293T cells results in cell toxicity and this effect can be used as a readout of the drug activity. Recently, a reporter-based assay for antiviral drug screening in human cell culture at BSL2 was proposed by Froggatt H.M. et al. [128]. The authors described a reporter based on green fluorescent protein that emits only after cleavage by M^pro^. The execution of the assay on living cells allows us to evaluate simultaneously the inhibition of the protease and the cell viability. Although this plasmid-based expression showed many advantages, it requires further screening tests in the context of coronavirus infection. Specifically, the effects of M^pro^ inhibitors correlated to cross-viral protein interactions would be missed. This approach was used to develop the 3CLglow commercial biosensor [129]. The assay involves two components, the M^pro^ enzyme BacMam and the M^pro^ biosensor BacMam; the protocol was optimized and validated on human embryonic kidney (HEK 293T) cells. The BacMam vector carrying the fluorescent biosensor in these assays is a modified baculovirus (Figure 24).

## 5. Cell-Based Assays to Screen Novel SARS-CoV-2 Antiviral Drugs

The evaluation of the ability of a drug candidate to prevent in vitro viral replication is typically carried out in cell-based assays. Specifically, the test on cell cultures infected with SARS-CoV-2 requires biosafety level 3 (BSL-3) facility and must be performed under appropriate conditions. An important aspect of the screening of compounds for the treatment of new viral diseases such as COVID-19, is represented by the choice of the bioassay and the selection of a suitable cell type.

To date, the in vitro antiviral effectiveness of potential COVID-19 inhibitors has been mainly evaluated by: (i) a SARS-CoV-2 CPE assay, which is a standard infectivity assay based on viral infection of host cells followed by visual monitoring of the virus-induced CytoPathic Effects (CPE); (ii) a SARS-CoV-2 VYR assay, which is a Virus Yield Reduction assay allowing a virus titer calculation and a EC_90_ determination; (iii) the qRT–PCR analysis, namely the traditional quantitative real-time polymerase chain reaction used to measure ongoing viral replication.

To carry out the SARS-CoV-2 CPE assay, different concentrations of tested compound are added to the plates to measure the effective concentration at which the drug inhibits viral CPE 50% (EC_50_) or, in the absence of virus, the inhibitory concentration at which growth is inhibited by 50% (IC_50_). Percent CPE inhibition is defined as [(test compound–virus control)/(cell control–virus control)] × 100. Percent cell viability is defined as (test compound/cell control) × 100. An active compound can be defined “hit” if it exhibits a % of CPE inhibition >50 without compromising cell viability [130].

The CPE assay relies on the visual observation on the damage of infected cells under a microscope. The observation of cell morphology allows to determine the degree of CPE compared to the protected monolayer cells. The evident limitation of this method is that of combining two types of measurement to evaluate the antiviral activity, i.e., the classical plaque reduction (viral plaques were quantified by visual inspection and compared to non-treated virus control) [20] and the loss of neutral red staining (NRS) uptake which corresponds to loss of cell viability [131]. Briefly, the CPE is quantified by neutral red dye uptake, exploiting the ability of viable cells to incorporate and bind the supravital dye neutral red in the lysosomes. NRS is a weak, cationic dye that readily penetrates cell membranes by diffusion, accumulating intracellularly in lysosomes. It provides a quantitative estimation of the cell viability since the amount of retained dye is proportional to the number of viable cells. The optical density is read on a spectrophotometer at 540 nm and the effective concentrations of tested compound required to prevent virus-induced CPE by 50% (EC_50_) and to cause 50% cell death in the absence of virus (CC_50_) are calculated [131]. As an example, Boceprevir has an EC_50_ of 1.90 μM against SARS-CoV-2 virus in the cellular viral CPE assay: EC_50_ was calculated based on virus-induced CPE quantified by neutral red dye uptake after 5 days of incubation [25].

Moreover, to further investigate the extensive cell death caused by apoptosis or necrosis in the CPE region, specific staining can be also performed with a cell apoptosis/necrosis detection kit (blue, green, red) (i.e., Abcam 176749) able to detect both apoptosis and necrosis at the same time [132]. It is noteworthy to highlight that when compounds are selected from a preliminary screening based on the inhibition of the SARS-CoV-2 induced CPE, we cannot exclude that they have no antiviral effect, but rather only protect the cells from CPE [77]. Therefore, the combination of multiple screening assays is generally preferred.

The SARS-CoV-2 Virus Yield reduction (VYR) assay is conducted by first replicating the virus in the presence of tested compounds. Host cells are seeded and grow overnight to confluence. A sample of the supernatant fluid from each tested concentration is collected on day 3 post infection and tested for virus titer. The concentration of compound required to reduce virus yield by 90% (EC_90_) is determined by regression analysis [131].

Currently qRT-PCR is the most precise method to measure gene expression and to quantify the viral genome. It can be readily applied to high-throughput screening and has many advantages over the assays based on subjective observation and laborious scoring of CPE or plaque formation. In a typical qRT–PCR analysis, the antiviral efficacy is evaluated by quantification of viral copy numbers in the cell supernatant. Briefly, host cells are pre-treated with the tested compounds, followed by virus infection. Viral RNA is extracted from the supernatant of the infected cells and quantified by RT-PCR. The 50% effective concentration (EC_50_, compound concentration required to inhibit viral RNA replication by 50%) is determined using logarithmic interpolation. For the evaluation of the CC_50_ (the concentration that reduces the total cell number by 50%), the same culture conditions are used as for the determination of the EC_50_, without addition of the virus, and cell viability is measured using a cell viability assay based on a fluorescent method (i.e., CellTiter Blue^®^, Promega) [77].

From literature, it emerged that a panel of laboratory cell lines can be used to cultivate SARS-CoV-2 [133]. The susceptibility of a number of cell lines to SARS-CoV-2, especially from human origin, can be fruitfully exploited to identify potential antiviral compounds, to investigate intracellular trafficking, to produce the virus in large quantities for vaccine research, and to develop incisive therapeutic approach.

For in vitro antiviral studies, different cellular systems have been employed so far to demonstrate the susceptibility of SARS-CoV-2 to a potential drug. Vero E6 cells (African green monkey kidney cells) are a well-known model system that produces high virus titers and displays visual CPE. Since 2003, Vero E6 cells have been extensively used for SARS-CoV research in cell culture-based infection models as they support viral replication to high titers, likely due to a high expression of ACE-2 receptor [134].

Recently, the use of Vero E6 cells, human epithelial pulmonary cell line (A549) and a SARS-CoV-2-infection model utilizing human primary monocytes has been reported to evaluate the efficacy of atazanavir (ATV) alone or in combination with ritonavir (RTV) on SARS-CoV-2 replication. Interestingly, experimental results pointed out that ATV showed a nearly 10-fold increase in potency for inhibiting SARS-CoV-2 replication in A549 (EC_50_ = 0.22 μM) compared to Vero E6 cells (EC_50_ = 2.0 μM). The observed efficacy against SARS-CoV-2 in A549 cells is likely to be consistent with its bioavailability in the lungs in experimental models [56].

Remdesivir, a drug that has faced a storm of controversy, was reported to inhibit SARS-CoV-2 in the VYR assay with an EC_50_ of 0.77 μM in Vero E6 cells; a similar value (0.651 μM) was determined by CPE in Vero E6 cells. In human lung cells and primary human airway epithelial cultures, an EC_50_ of 0.01 μM was calculated by qRT-PCR, whereas weaker activity was observed in Vero E6 cells (EC_50_ = 1.65 μM) due to their low capacity to metabolize the drug [135]. Overall, the antiviral activity seems to depend to a great extent on the cell type used in the screening [12]. The differences in EC_50_ could be partially explained by intrinsic differences of quantification methods and assay conditions, such as incubation period and virus input. Moreover, the metabolic capacity of host cells should be also taken into account, especially for testing drugs requiring a bio-activation, i.e., nucleoside analogues, such as remdesivir, that undergo intracellular conversion to the active metabolite triphosphate [135]. Although Vero E6 cells support robust replication of SARS-CoV-2, extreme caution should be exercised in interpreting drug efficacy and potency measured in Vero E6 cells [135]. In fact, since the metabolism of remdesivir in Vero E6 cells appears not very efficient, they might not be the appropriate cell type to investigate the antiviral efficacy of nucleotide prodrug-based antivirals. Moreover, the selection of the cell line strictly influences the choice the bio-assay to be carried out and also its practical execution: as an example, Vero E6 cells support replication of SARS-CoV-2 and produce CPE 48 h after SARS-CoV-2 infection, whereas two human cells lines, HEP-2 (epithelial cell line form lagyngeal carcinoma) and Caco-2 (epithelial cells from colorectal adenocarcinoma) supported virus replication, but only HEP-2 cell lines produce CPE 120 h after inoculation, while Caco-2 showed only discrete modification as compared to control but no real CPE [133].

Altogether, these findings emphasize the need for careful selection of multiple cell types and combined methods to study the effectiveness of potential antiviral drug to fight COVID-19. Despite considerable efforts worldwide and over 300 active clinical evaluations, no effective countermeasure exists to combat the pandemic so far.

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
