# Peer review of "SARS-CoV-2 M^pro^: A Potential Target for Peptidomimetics and Small-Molecule Inhibitors"

_biomolecules, 2021, doi:10.3390/biom11040607_

Round 1

Reviewer 1 Report

This manuscript is an important and comprehensive review of peptidomimetics, and small-molecule inhibitors of SARS-CoV-2 Mpro with detailed analysis of the repurposed drugs and the newly synthesized molecules developed up to date for the treatment of COVID-19 and also overview of the screening assays.

It would be worthy of adding a brief note of the following issues:

1.The following facts about remdesivir:

- Remdesivir Is First and Only FDA-Approved Treatment for COVID-19 in the US https://www.gilead.com/news-and-press/press-room/press-releases/2020/10/us-food-and-drug-administration-approves-gileads-antiviral-veklury-remdesivir-for-treatment-of-covid19

- Remdesivir has been approved or authorized for temporary use as a COVID-19 treatment in approximately 50 countries worldwide. https://www.gilead.com/Purpose/Advancing-Global-Health/COVID-19/Veklury-global-marketing-authorization

- New results from the comparative effectiveness research study of adults hospitalized with COVID-19, receipt of remdesivir was associated with faster clinical improvement

https://jamanetwork.com/journals/jamanetworkopen/fullarticle/2777863

2.The information regarding a first-in-class cyclic peptide inhibitor against the SARS-CoV-main protease (Mpro) could be added

 https://www.biorxiv.org/content/10.1101/2020.08.03.234872v1.full

Author Response

 Reviewer#1

Comments and Suggestions for Authors

This manuscript is an important and comprehensive review of peptidomimetics, and small-molecule inhibitors of SARS-CoV-2 Mpro with detailed analysis of the repurposed drugs and the newly synthesized molecules developed up to date for the treatment of COVID-19 and also overview of the screening assays.

It would be worthy of adding a brief note of the following issues:

1.The following facts about remdesivir:

- Remdesivir Is First and Only FDA-Approved Treatment for COVID-19 in the US https://www.gilead.com/news-and-press/press-room/press-releases/2020/10/us-food-and-drug-administration-approves-gileads-antiviral-veklury-remdesivir-for-treatment-of-covid19

- Remdesivir has been approved or authorized for temporary use as a COVID-19 treatment in approximately 50 countries worldwide. https://www.gilead.com/Purpose/Advancing-Global-Health/COVID-19/Veklury-global-marketing-authorization

- New results from the comparative effectiveness research study of adults hospitalized with COVID-19, receipt of remdesivir was associated with faster clinical improvement

https://jamanetwork.com/journals/jamanetworkopen/fullarticle/2777863

RE to 1: We thank the reviewer for his/her positive feedback to our review article. As suggested, the statements about the antiviral drug remdesivir have been added in the main text, including its current approval status in US and throughout the world. New results of comparative effectiveness research study have been mentioned and properly referred as well. Revisions are marked in red and highlighted in yellow in the new manuscript for easy check.      

  1. The information regarding a first-in-class cyclic peptide inhibitor against the SARS-CoV-main protease (Mpro) could be added

 https://www.biorxiv.org/content/10.1101/2020.08.03.234872v1.full

RE to 2: As suggested, information regarding the first-in-class cyclic peptide inhibitor of SARS-CoV-2 Mpro has been added. We actually provided an entire paragraph (i.e. 3.2.7) dealing with it and the related paper.

Reviewer 2 Report

SARS-CoV-2 Mpro: A potential target for peptidomimetics and small-molecule inhibitors

In the present review article, the authors offered a detailed analysis of the repurposed drugs and the newly synthesized molecules developed to date for the treatment of COVID-19 which share the common feature of targeting SARS-CoV-2 Mpro, as well as a brief overview of the main enzymatic and cell-based assays to efficaciously screen such compounds. In my opinion, the study is interesting and innovative. However, I have some comments:

Comment (1): Some parts of the text are confusing and needs a thorough revision of English.

Comment (2): Abstract. The background topic is poor. I recommend to including some sentences about SARS-CoV-2 Mpro as this is the main subject of the review.

Comment (3): Introduction. There is a brief review of existing knowledge and relevance of study.

- Line 41: add ref.

Comment (4): Figures and legends. Legends are adequate and figures are necessary to understand the results obtained.

Author Response

Reviewer#2

Comments and Suggestions for Authors

SARS-CoV-2 Mpro: A potential target for peptidomimetics and small-molecule inhibitors

In the present review article, the authors offered a detailed analysis of the repurposed drugs and the newly synthesized molecules developed to date for the treatment of COVID-19 which share the common feature of targeting SARS-CoV-2 Mpro, as well as a brief overview of the main enzymatic and cell-based assays to efficaciously screen such compounds. In my opinion, the study is interesting and innovative. However, I have some comments:

RE: We thank the reviewer for his/her positive feedback to our review article. Hereinafter there are the answers to his/her comments. Revisions are marked in red and highlighted in yellow in the new manuscript for easy check.   

Comment (1): Some parts of the text are confusing and needs a thorough revision of English.

RE to 1: As indicated by the reviewer, some parts of the text (especially those ones pertaining non-peptidic inhibitors, enzymatic assays and cell-based assays) have been thoroughly revised

Comment (2): Abstract. The background topic is poor. I recommend to including some sentences about SARS-CoV-2 Mpro as this is the main subject of the review.

RE to 2: According to the reviewer’s observation, the abstract has been added with some sentences pertaining the topic of the review article.

Comment (3): Introduction. There is a brief review of existing knowledge and relevance of study.

- Line 41: add ref.

RE: Reference has been added accordingly.

Comment (4): Figures and legends. Legends are adequate and figures are necessary to understand the results obtained.

RE: We appreciate the comment. In the revised version of the manuscript, Figures are re-numbered and/or revised on the basis of requests and updates.